# GEOMETRIC MULTIGRID NEURAL NETWORKS

## ABSTRACT

We introduce Geometric Multigrid Neural Networks (GMNN), a novel network structure for geometric deep learning on point clouds and surfaces. Convolutional neural networks face a common challenge: how can relevant features be communicated over longer distances? Our architecture facilitates long-distance communication with Geometric Multigrid Convolution (GMC) blocks, which apply convolutions in parallel to features defined on each level of a multigrid representation of the surface, and enable communication all the way up and down the hierarchy. We observe two major structural advantages of such a network: First, because each GMC operates on all levels of the multigrid hierarchy, even early stages can make use of coarse-scale information and receptive field grows rapidly with depth. Second, networks built with this backbone have the freedom to route information between different scales, including in ways not possible for other architectures. Because of these advantages, we find that a GMNN can combine the fast training of a shallow network with the greater expressiveness of a deeper, larger network. We build a GMNN from the components of a state-of-the-art U-Net, and find that on real tasks it can match or exceed the accuracy of the base network while using fewer epochs and roughly half the parameter count.

## 1 INTRODUCTION

Advances in 3D capture and modeling technologies have made geometric data increasingly accessible, with applications spanning from computer graphics and medical imaging to engineering and manufacturing. As a result, the analysis and processing of geometric data have become key problems, with advances benefiting a wide range of use cases. Over the past decade, successful deep learning techniques from the image and language domains have been adapted to geometric data. These geometric deep learning methods have achieved breakthroughs for various challenging problems in 3D data processing and analysis.

Convolutional neural networks improve efficiency and introduce useful priors by using local operators. However, this localization comes with limitations: multiple layers are required to integrate information across distant regions of the domain and to capture coarse-scale features. Several strategies have been proposed to address this, including augmenting networks with non-local, low-frequency Laplace eigenfunctions (Sharp et al., 2022), introducing additional non-local connections based on feature-space proximity (Wang et al., 2019), and applying convolutions to progressively coarser point clouds (Qi et al., 2017). Despite these advances, limitations remain on the efficient extraction and combination of features at different spatial locations and across scales.

We introduce a novel architecture: the Geometric Multigrid Neural Network (GMNN, fig. 1). It represents features in each layer using a multigrid representation of the domain. By storing features on different levels of the multigrid hierarchy, the network can efficiently represent information at multiple scales. Layers are connected through Geometric Multigrid Convolution (GMC) blocks, which perform convolutions within the same level of the multigrid hierarchy and between different levels, enabling effective extraction and integration of multiscale features. GMNNs are constructed by chaining GMC blocks in series.

This architecture equips the network with flexible mechanisms for multi-scale feature extraction and integration, enabling appropriate organization of information flow for the task at hand. To illustrate how it differs from other architectures, consider the receptive field: since each GMC block links features in each level to features on coarser levels of the multigrid hierarchy in the preceding layer,

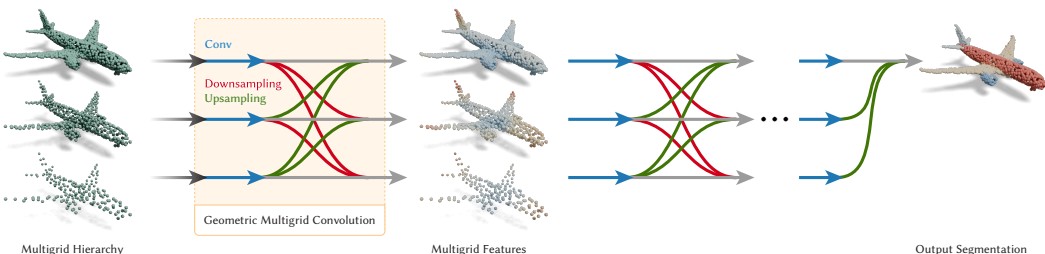

Figure 1: Geometric Multigrid Convolutions operate on features defined on the levels of a multigrid. This allows Geometric Multigrid Neural Networks to extract features at a range of scales, starting at the earliest layers, and to efficiently fuse information both across scales and between spatially distant regions on the surface.

the receptive field in a GMNN expands rapidly. We quantify this effect in our experiments and compare GMNN against other architectures.

Comparing GMNN with alternative architectures such as U-Net (Ronneberger et al., 2015) high-lights fundamental differences. In a U-Net, information follows a fixed trajectory: features are first extracted at fine scales and then progressively aggregated at coarser levels. In contrast, GMNN offers greater flexibility. It can follow the same fine-to-coarse pathway, but it can also exploit alternative routes, such as coarse-to-fine or paths that traverse up and down the multigrid hierarchy multiple times. This flexibility translates into greater expressive power for GMNN, which we demonstrate on a synthetic function approximation task. In comparisons on benchmark segmentation tasks, GMNN matches or surpasses the accuracy of state-of-the-art models, while converging in fewer epochs and using roughly half as many parameters.

## 2    RELATED WORK

Here, we provide a summary of existing architectural solutions to enable long-range communication in convolutional neural networks, starting on other domains, and then on meshes and point clouds.

**Multiscale architectures for images and non-geometric graphs**    The U-Net, first introduced in Ronneberger et al. (2015), enables segmentation of large images with fewer convolutions by oper-ating on progressively coarser (lower resolution) domains before expanding to progressively finer domains. Multigrid Neural Architectures (Ke et al., 2016) demonstrated the potential of multi-grid convolutions on the image domain. The authors augment well-known architectures with multi-grid convolutional layers and demonstrate accuracy improvements on common segmentation bench-marks while retaining efficiency in parameters and compute. Feature Pyramid Networks (Lin et al., 2017) (FPN) construct features on multiple scales and apply convolutions to those scales; features from all scales are merged at the end of the network before producing labels. UNet++ (Zhou et al., 2018) extends the U-Net architecture by keeping the convolutions in higher resolutions, where the original U-Net would downsample and only continue in the lower resolution, proverbially 'filling in' the U-shape. They also connect adjacent scales in each layer. Multigrid Graph Neural Net-works (Taghibakhshi et al., 2023) (MG-GNN) adapts a limited multigrid architecture to the graph learning domain, and shows that a two-level multigrid structure has benefits for graph-partitioning tasks. Closest to our approach, IM-MPNN (Finder et al., 2025) implements a more complete multi-grid architecture for graph learning, with 4 scales and communication between adjacent scales. It confirms advantages on large-diameter graph problems, where communication across the original graph requires many convolutions. Each of these approaches show that variations on the U-Net structure can benefit deep learning on other domains; our architecture incorporates some of these ideas for use on point clouds.

**U-Nets on point clouds**    The structure of PointNet++ (Qi et al., 2017), a pioneering method for learning on point clouds, resembles that of a U-Net (Ronneberger et al., 2015). In the 'encoder,' the features in the point cloud are progressively downsampled with farthest point sampling (FPS) fol-lowed by max-pooling in a local neighborhood (KNN or ball query). The 'decoder' then upsamples

the features with linear interpolation. Features from the encoder are connected to the decoder using skip-connections between corresponding scales. This base structure has been adapted and developed by many follow-up works. Among many others, PointCNN (Li et al., 2018), KPConv (Thomas et al., 2019), KPConvX (Thomas et al., 2024), PointTransformer v2 (Wu et al., 2022) and v3 (Wu et al., 2024) also employ U-Net architectures. The main contributions of these works lie in improving the convolution operator, e.g., with a graph- or point-based variant of convolution or attention (transformer). While the works also change the architecture, e.g., by adding residual connections, they do not fundamentally alter the U-Net structure. More recently, PointNeXT (Qian et al., 2022) and DeLA (Yang et al., 2024), show that applying these changes in architecture and training procedure with PointNet-style convolutions can lead to state-of-the-art results. Our work is orthogonal to the contributions on the convolution operators, as we explore different ways to connect scales in the hierarchy, regardless of the operations within the scales. Therefore, we employ simple building blocks, such as the MLPs and max-aggregation used in PointNet++ and DeLA. In practice, the insights from our work could be combined with different blocks such as graph-based, convolution-like or transformer-style layers in a neural network for point clouds.

**Alternative architectures on Point Clouds**    On point clouds, works like DGCNN (Wang et al., 2019) and DiffusionNet (Sharp et al., 2022) have explored non-architectural ways to connect information at longer ranges, but fewer works have explored alternatives to the U-Net. Multiresolution Tree Networks (Gadelha et al., 2018) (MTN) adopt a multigrid architecture (Ke et al., 2016) for point clouds. Rather than working natively in 3D, they use spatial sorting to represent point clouds as a 1D structure and apply a 1D convolution on the point clouds. They maintain 3 scales of a multigrid through much of the network, but eventually pool all scales to a global node (to enable use as an auto-encoder). PointHR (Qiu et al., 2023) resembles the augmentation of UNet++, 'filling in' convolutions in the higher resolutions throughout the network. Our architecture goes beyond both MTN and PointHR by performing convolutions on all resolutions throughout the network; at no point does it operate exclusively on coarse scales (as MTN does at the end of its encoder) or exclusively at fine scales (as PointHR does at the beginning of its encoder).

## 3    METHOD

A GMNN is designed to learn to communicate information between finer and coarser scales as needed, rather than following a predefined path. This is achieved with a multigrid feature representation and a geometric multigrid convolution block. We define generic versions of these in this section, with sampling procedures, convolution operators, and other components left unspecified. In section 4, we instantiate GMNNs for each task by adapting components from other networks.

### 3.1    MULTIGRID FEATURES

Throughout a GMNN, features are defined over a 'multigrid' representation of the input domain. Given an input point cloud $P^1 \in \mathbb{R}^{N \times 3}$ and a desired number of levels $S$, we construct a sampling hierarchy of point clouds, $\{P^1, \dots, P^s, \dots P^S\}$, with point counts $N^s$. Each sampling $P^s$ represents one level of the multigrid hierarchy. The point cloud hierarchy is associated with multigrid features: a set of $C$ features $X^s \in \mathbb{R}^{N^s \times C}$ per level $P^s$ .

### 3.2    GEOMETRIC MULTIGRID CONVOLUTION BLOCKS

A Geometric Multigrid Convolution (GMC) Block enables multigrid connectivity by composing convolutions with upsampling and downsampling stages to connect different levels of the multigrid hierarchy.

**Parallel convolutions**    Point cloud convolutions, such as PointNet (Qi et al., 2017), are applied to each level in parallel to produce a new set of multigrid features $X^{s'}$, as shown in fig. 2a. These convolutions have independent parameters and are not shared, because we expect that different types of features can be found at different levels.

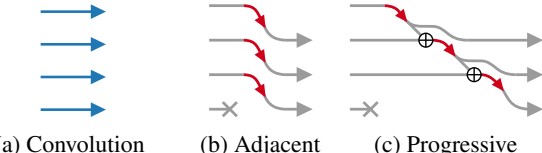

(a) Convolution      (b) Adjacent      (c) Progressive

Figure 2: Multigrid operations on 4 scales. Parallel convolution (left), adjacent and progressive downsampling (right). Upsampling operations are the inverse of downsampling.

**Transfers between scales**  We can use down- and upsampling to match spatial resolutions and communicate the new features $X^{s'}$ to the adjacent levels $s + 1$ or $s - 1$ (fig. 2b).

We aim to design a transfer block that can communicate between each pair of multigrid levels. Naively, this would require $S(S - 1)/2$ upsampling operations and $S(S - 1)/2$ downsampling operations. Instead, we achieve this universal connectivity more efficiently by performing transfers progressively, rather than in parallel. This reduces our requirement to $S - 1$ upsampling operations and $S - 1$ downsampling operations (fig. 2c). Progressive prolongation produces new features on all levels except for the coarsest level ($S$), restriction on all except for the finest level (1). There are many options for incorporating the level-transferred features before the next stage. In our implementations, we use a linear layer to match dimensionality and add them to the features of each level.

**Transfer layouts**  Given components for communicating features within each level (parallel convolutions) and between levels (progressive upsampling, downsampling), there is a degree of freedom in how we arrange the connections. We explored arrangements which alternate between convolutions and transfers, and found that a sufficiently deep GMNN is not strongly sensitive to their ordering. For most tasks, we use a GMC block that first performs progressive downsampling, then upsampling, and finally applies parallel convolutions.

### 3.3 GEOMETRIC MULTIGRID NEURAL NETWORKS

A GMNN is made up of $D$ multigrid convolution layers, each with $S$ levels, placed in series (fig. 1). Because the neighborhood queries used in convolutions and pooling operations will be used in every block of the network, we precompute the point hierarchy and neighborhoods once and reuse them throughout. A GMC block expects input features on all levels. For the first layer, this can be achieved by extracting local shape information independently on each level or by progressively downsampling the finest-level features $X^1$ from the input point cloud $P^1$ to generate representations for all coarser levels.

For segmentation, progressive upsampling is applied at the end of the network, ensuring features learned on all levels contribute to the final output on the finest level. After this, a conventional segmentation head (point-wise MLP) is used. Classification can be done with the reverse of this approach — ending with progressive downsampling to collect all features to one node per point cloud, and then applying an MLP-based classification head to the features of that point.

### 3.4 PROPERTIES

The GMNN architecture allows information to be represented at multiple scales within the multigrid hierarchy of each layer, with GMC blocks extracting and combining features across these scales. Unlike other networks, where information flow is prescribed—such as the fine-to-coarse progression in U-Nets—GMNN can flexibly organize information flow to suit the learning task. We discuss two major advantages of a GMNN over other architectures: First, its convolutions have larger receptive fields, stemming from an ability to communicate information across long distances with fewer intermediate steps (section 3.4.1). Second, a GMNN offers greater expressiveness, arising from its ability to route information flexibly between levels (section 3.4.2).

Similar benefits could be obtained by building a deeper conventional model, however, an overlong gradient path can increase training time and exacerbate problems such as vanishing gradients. By

arranging its convolutions in parallel, a GMNN obtains some advantages of deeper networks while avoiding these drawbacks.

### 3.4.1 Communication Advantage

The first stage of a U-Net allows information to travel only along the finest level of multigrid hierarchy, but the first stage of a GMNN can propagate signals much further using the coarser levels. In section 4.1, we measure this property for several model architectures and show the advantages of a GMNN. In section 4.2 and appendix A.1, we find that this greater connectivity allows the network to converge quickly. This difference is intuitive because a U-Net must learn to carry useful information on the fine scales deeper into the network in order to propagate it further, where in a GMNN each layer after the first has direct access to information from longer distances.

### 3.4.2 Expressiveness Advantage

A U-Net has a prescribed path for how information must flow through a network: first between fine-level nodes, then progressively coarser. In contrast, a GMNN allows for the free exchange of information between levels. A sufficiently deep GMNN is a superset of a U-Net; it contains the fine-coarse-fine route, but also more complex routes, including the ability to make multiple trips from fine to coarse and back. Because of this, a GMNN can organize the communication of features between scales according to the specific characteristics of the learning target. We find that this translates to greater expressiveness across tasks. In section 4.3, we show how this translates to higher accuracy on a difficult synthetic function approximation task. In section 4.4, we find that on real segmentation tasks, our architecture allows us to compete with state-of-the-art models using only half the parameters. We note that in our experiments the GMNN tends to achieve much higher accuracy on the training set than the base U-Net, even when its advantage on the validation set is small. This indicates possible overfitting, which is expected of a more expressive model. This could be specific to factors like the complexity of the task or the size of the dataset. Mitigating this with standard regularization techniques could further improve validation accuracy.

## 4 Experiments

When comparing architectures, we consider three types of networks: a flat network, which simply chains convolutions on the input (finest) level; a U-Net, which is a typical depth-scaled design, applying blocks of convolutions to each level in series from finest to coarsest, followed by a simple MLP-based upsampling stage; and GMNN, which chains GMC blocks, described in the previous section. Unless otherwise stated, these networks all use PointNet++-style convolutions. U-Net and GMNN networks are used with identical hierarchy-construction procedures, so the message-passing is done over equivalent nodes and edges on each level.

### 4.1 Receptive Field

We expect that early access to coarse levels provides GMNN with a structural advantage in expanding its receptive field. This advantage is quantified by evaluating the receptive fields of different convolutions within a network. Receptive fields are estimated by tracking how information propagates from a single starting node across the domain. As a test case, we use the Stanford bunny mesh, chosen for its relatively low connectivity, with convolutions applied over the original mesh edges. Coarser levels are generated using 50% FPS sampling. Downsampling relies on edges from the fine scale, while upsampling connections are formed via nearest-neighbor clustering. Convolutions operate on the connectivity between clusters. Starting from node 0, we record the depth at which each node becomes reachable.

Figure 3 illustrates how the receptive field grows with network depth for different architectures. In the flat network, communication is restricted to the finest scale, so the receptive field remains small. The U-Net performs better; while its early convolutions behave like those of the flat network, later stages operate on coarser meshes and therefore reach further. In contrast, the GMNN achieves rapid growth in receptive field even at shallow depths, since its early stages already include convolutions on coarse scales. The accompanying images show the receptive field of a convolution at the green-marked node after 8 stages. In the flat network, 8 fine-scale convolutions cover only a limited range.

In the U-Net, 4 fine scales and 4 secondary scales moderately extend the range. In the GMNN, 8 GMC blocks — each operating on all scales — enable long-range communication.

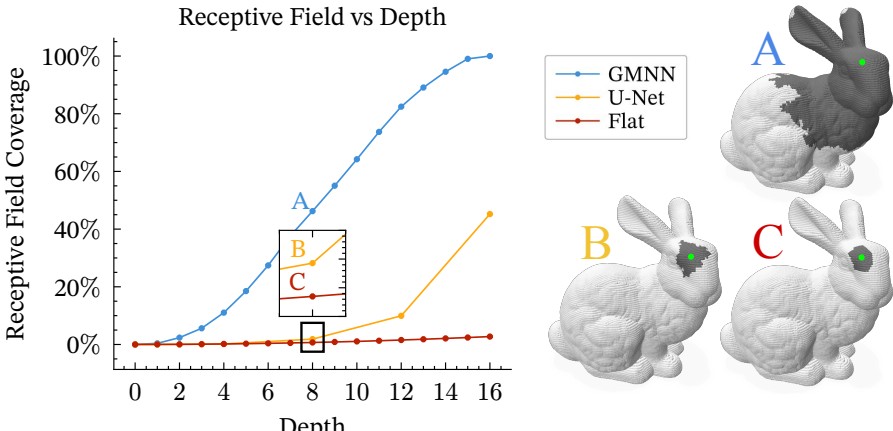

Figure 3: Left, receptive field of models at different depths. Right, receptive fields at a depth of 8; query node marked in green, contributing nodes in black.

## 4.2 FAST CONVERGENCE

We expect that broader access to non-local information in a GMNN allows it to more readily discover useful ways features from different parts of the domain can be combined. In practice, this implies that a GMNN can achieve higher accuracy in fewer epochs than comparable architectures. This trend is evident in the early stages of long training runs (see appendix A.1), and is even more pronounced under accelerated training schedules.

Table 1: ShapeNet, 10 epochs.

| Model | Ins. IoU | Cat. IoU |
|---|---|---|
| Flat | $82.1 \pm 0.1$ | $77.8 \pm 0.1$ |
| U-Net | $84.5 \pm 0.1$ | $80.2 \pm 0.2$ |
| GMNN | $\mathbf{84.7 \pm 0.1}$ | $\mathbf{81.3 \pm 0.3}$ |

Table 2: S3DIS, 10 epochs.

| Model | mIoU | Cls. Acc | OA |
|---|---|---|---|
| Flat | *(Omitted, high memory requirements)* | | |
| U-Net | $39.0 \pm 0.3$ | $46.5 \pm 0.4$ | $78.0 \pm 0.3$ |
| GMNN | $\mathbf{63.0 \pm 0.7}$ | $\mathbf{71.2 \pm 0.2}$ | $\mathbf{88.0 \pm 0.3}$ |

Parameter-count-matched networks with different backbones. Other hyperparameters are identical between models. 3 trials were used for each run.

We build several models of different architectures, each with exactly 16 PointNet++-style convolutions (the GMNN contains 4 GMC blocks) and channel counts chosen so that all have approximately 0.9M parameters. We use a similar training procedure to DeLA and PointNeXT, except for the learning rate schedule which is compressed to 10 epochs, a fraction of the normal training time. We find that on ShapeNet (table 1), both the U-Net and GMNN converge quickly on common classes, but the GMNN converges to a much higher accuracy on less common classes (Cat. IoU). On the larger point clouds of S3DIS (table 2), where information must be communicated further, this gap widens. The U-Net struggles to reach high accuracy in such a short training cycle. GMNN reliably reaches 10% higher overall accuracy (OA) than the U-Net, with even wider margins on uncommon classes (mIoU, Cls. Acc).

## 4.3 EXPRESSIVENESS

We expect that the additional flexibility of our architecture allows for increased expressiveness. That is, given similar parameter counts, a GMNN can express more complex functionality than a flat network or a U-Net. We can evaluate this flexibility with a difficult function approximation task: Given a large mesh of a crumpled ball of paper, we train models to map points to their locations on the original flat sheet (fig. 4). Relative point positions $(p_j - p_i)$ are provided by the PointNet++-style convolutions to ensure that the network does not simply learn a mapping on the embedding space, but learns from local surface features.

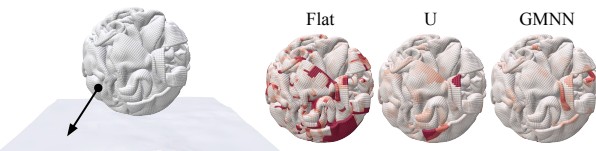

Table 3: Paper Un-crumpling Task, after 10,000 training iterations.

| Model | MAE | MSE |
|-------|-----|-----|
| Flat | 1.69 ± 0.21 | 20.4 ± 1.6 |
| U-Net | 0.67 ± 0.14 | 2.0 ± 1.2 |
| GMNN | **0.45 ± 0.02** | **0.9 ± 0.1** |

Figure 4: Paper un-crumpling task (left). Localization of error for different architectures (right).

We adapt the architectures from section 4.2, configured so that each has just over 0.5M parameters. The dataset consists of a single mesh of approximately 150,000 vertices (IndefinGaming (2021)). Convolutions are performed using neighborhoods based on edges of the original mesh, and for the U-Net and GMNN the hierarchy is produced using a similar approach to section 4.1. In table 3 we find that switching from a U-Net to a GMNN reduces absolute error by over 30% vs. the U-Net, and squared error by more than a factor of two.

## 4.4 COMPARISONS

Our goal is to test the effect of multigrid connections when augmenting existing networks. To simplify the analysis, we start from a PointNet-style (MLP followed by maximum aggregation) U-Net. In our comparisons against state-of-the-art, we use DeLA (Yang et al., 2024) as the base network, replicated in our own codebase, and augment it by placing its convolutions into a series of GMC layers. We denote the DeLA architecture as U-Net and our augmented variant as GMNN.

The original U-Net backbone for DeLA part-segmentation is shown in fig. 5a. It uses a single set of PointNet-style convolutions at the start of the network to embed spatial information (yellow arrows). This information is passed through convolutions (blue) containing efficient 'decoupled' local aggregations, downsampling (red), and a spatial regularization step (yellow dots) computes a loss encouraging the network to preserve spatial information throughout. Finally, point-wise MLPs convert the features on all scales to the same channel count, so that a progressive upsampling step can add all outputs before the segmentation head. For part-segmentation, category labels are also provided as features on a global node.

**ShapeNet-Part Segmentation** To assess part segmentation performance, we use the ShapeNet-Part dataset (Wu et al., 2015), a common benchmark that consists of 13 categories of clean object point clouds (e.g. plane, chair) divided into 50 total parts (e.g. wing, seat). As input, the model takes point coordinates, normals, and category labels (provided just before the segmentation head). As output, the model produces logits indicating the associated part for each point. We configure GMNN with the same number of channels as DeLA on the finest scale, and set the number of channels on the other scales to produce a small version of the model and a wider version with more parameters. Both models have a depth of 4 GMC layers, with half as many local aggregations per point convolution, so that the effective model depth is shallower than that of DeLA. We find that this task is prone to overfitting with deep networks. To mitigate, we make GMNN shallower without reducing parameter count by placing upsampling and downsampling blocks in parallel. For fair comparison, we use the same sampling procedures, dataset regularization, train/test split, and

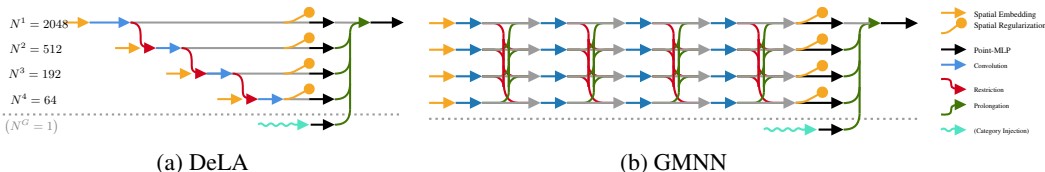

(a) DeLA        (b) GMNN

Figure 5: Architecture comparison. The part-segmentation variant of DeLA (U-Net) and the same convolutions incorporated into a GMNN. Each arrow represents an operation with input and output features (channel counts left unspecified) on the points of the network. Where lines meet, features are added as residuals.

Table 4: Segmentation results on ShapeNet-Part.

| Method | Ins. IoU | Cat. IoU | Params. |
|---|---|---|---|
| PointNet++ (Qi et al., 2017) | 85.1 | 81.9 | 1.0 M |
| PointNeXT-S (C=64) (Qian et al., 2022) | 86.9 | 84.8 | 3.7 M |
| PointNeXT-S (C=160) (Qian et al., 2022) | 87.0 | 85.2 | 22.5 M |
| PTv1 (Zhao et al., 2021) | 86.6 | 83.7 | 7.8 M |
| SPoTr (Park et al., 2023) | 87.2 | 85.4 | 1.7 M |
| AVS-Net (Zhang et al., 2023) | 87.3 | 85.7 | - |
| PointHR (Qiu et al., 2023) | 87.2 | - | 7.4 M |
| DeLA (Yang et al., 2024) | 87.5 | **86.0** | 7.0 M |
| GMNN | 87.6 | 85.8 | 3.0 M |
| GMNN (Scaled) | **87.7** | **86.0** | 8.1 M |

Table 5: Segmentation results on S3DIS Area 5,
++ indicates pretraining on extra data was used.

| Method | | mIoU | Cls. Acc | OA | Params. |
|---|---|---|---|---|---|
| PTv2 Wu et al. (2022) | ++ | 72.7 | 78.0 | 91.6 | 12.8 M |
| PTv3 Wu et al. (2024) | ++ | 74.3 | 80.1 | 92.0 | 124.8 M |
| PTv3+Sonata Wu et al. (2025) | ++ | 76.0 | 81.6 | 93.0 | 124.8 M |
| PointNet++ Qi et al. (2017) | | 53.5 | - | 83.0 | **1.0 M** |
| PointNeXT Qian et al. (2022) | | 71.1 | 77.2 | 91.0 | 41.6 M |
| KPConv Thomas et al. (2019) | | 67.1 | 72.8 | - | 14.9 M |
| PTv1 Zhao et al. (2021) | | 70.4 | 76.5 | 90.8 | 4.9 M |
| PTv2 Wu et al. (2022) | | 71.6 | 77.9 | 91.1 | 12.8 M |
| PTv3 Wu et al. (2024) | | 73.4 | 78.9 | 91.7 | 124.8 M |
| PointHR Qiu et al. (2023) | | 73.2 | 78.7 | 91.8 | - |
| DeLA Yang et al. (2024) | | 74.1 | 80.0 | **92.2** | 7.0 M |
| GMNN | | **74.4** | **80.8** | 92.1 | 3.4 M |

postprocessing techniques as used by PointNeXT and DeLA. The training cycle is reduced to 150 epochs (vs. 250), with the learning rate schedule adjusted to match.

Table 4 places the performance of our adapted models in context with the original DeLA and other state-of-the-art approaches. We find that the smaller multigrid model matches DeLA with half the parameters. Scaling the model allows it to exceed DeLA on common categories. Both versions of the model also outperform state-of-the-art approaches, including modern transformers like SPoTr, AVS-Net, and PointHR.

**S3DIS Semantic Segmentation** We measure point cloud semantic segmentation performance on S3DIS (Armeni et al., 2016), a commonly used indoor-segmentation benchmark. It consists of high-resolution scans of rooms from 6 different parts of the Stanford offices. In line with common practice, we measure accuracy on Area 5. The trained model takes positions and RGB colors for each point and produces logits assigning the points to one of 13 classes (e.g., wall, chair, door). For this task, we configure a version of GMNN with the same base channels as DeLA and six shallow GMC layers, again with half as many local aggregations to create a shallower model. We keep the same training procedures as DeLA and PointNeXT, including dataset regularization, point cloud voxelization (with interpolation), and training scene cropping. The training schedule is again shortened, from 100 epochs to 75.

Table 5 places our results in context. We see that with a narrower, shallower network for half the total parameters, we match the accuracy of DeLA (OA) and significantly outperform it on uncommon classes (mIoU, Cls. Acc). Our network's performance improves over all state-of-the-art transformer models, given the same training data, with substantially fewer (1/35th) parameters. We include the results for PTv2 and PTv3 with pretraining for reference, but note that these results should not be directly compared to the other methods in the table, as they include extra training data.

## 4.5 TRANSFER CONNECTIVITY ABLATION

A novel aspect of GMNN is its use of progressive up- and downsampling stages to enable full connectivity between scales, as shown in fig. 2c. We can isolate the advantage of this by training networks with connections only between adjacent scales, as in fig. 2b. For reference, we also compare against networks that have no communication between scales, aside from the final pooling stage. This is similar to a feature pyramid network (Lin et al., 2017).

Table 6: Grid-transfer connectivity ablation on S3DIS, 75 epochs.

| Connectivity | mIoU |
|---|---|
| None | $71.0 \pm 0.1$ |
| Adjacent | $72.3 \pm 0.1$ |
| Progressive | $\mathbf{72.6 \pm 0.2}$ |

We set up an experiment on S3DIS with modified versions of a GMNN using DeLA convolutions. Table 6 shows how the connectivity between scales affects accuracy. We find that connections between scales are critical for producing an accurate network. Using progressive connections all the way up and down the network adds a consistent improvement over adjacent connections in mIoU at negligible additional compute or parameter cost.

## 4.6 TRAINING SPEED

A GMNN configuration may use more convolutions than competing architectures, like U-Net, especially on the finest scale. Edges can be precomputed and convolutions are performed in parallel, reducing runtime, but some additional cost remains. On S3DIS, we measure the performance of our best network against a re-implementation of DeLA (using the same convolutions) and find that each epoch takes 62% longer (173 vs. 281s). On the smaller meshes of ShapeNet, we find the difference narrows to 42% (59 vs. 82s). In both cases, the difference is partially compensated by the accelerated training schedule, making the GMNN 14% and 20% slower to train, respectively.

## 5 CONCLUSION

In this work, we introduce Geometric Multigrid Neural Networks (GMNN), a novel architecture that addresses key limitations in existing architectures for learning on point clouds. By operating on novel multigrid features with multigrid convolutional blocks, GMNN enables the representation and integration of features across multiple spatial scales within each layer, facilitating early coarse-scale processing and efficient information exchange across the point cloud. We showed that GMNN can be constructed from common components of other architectures and demonstrated its effectiveness through experiments. Our results show consistent improvements in performance, reduced parameter counts, faster convergence during training, and more efficient information propagation compared to state-of-the-art architectures such as U-Nets.

We view GMNN as a fundamentally novel framework for neural network design on point clouds. As our paper aims to isolate the advantages that come from an architectural change, several avenues for future improvement have not yet been explored. In particular, our work focuses on PointNet++-like convolutions, but the attentional convolutions of patch-transformer models like PTv3 could also be used with our backbone. Training technique changes like the use of pretraining could also be adopted from the transformer lineage, and may be appropriate because of the expressiveness of our model.

As our network is a superset of other architectures, we see potential for it as a tool for further design exploration. With targeted ablations, it may be possible to determine which connections are the most useful. Given this information, the GMNN could be 'pruned', leaving the necessary subset to create a bespoke architecture for a given task. Beyond producing a better architecture, this could give us more fundamental insights into how networks learn to route information.

**Reproducibility Statement**   A complete implementation of our method will be made available upon publication. Model configurations and scripts for running different experiments will be included alongside the implementation.

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

# A  APPENDIX

## A.1  EARLY SATURATION

We find that during training, our multigrid adaptation of DeLA tends to reach high accuracy on common classes much faster than its U-Net counterpart. This difference is clearest for instance-weighted accuracy metrics. Figure 6 shows relevant metrics for our implementations of GMNN and DeLA over the course of a training run, with identical training procedures. The multigrid model converges more quickly to high accuracy in both tests, and the accuracy of the U-Net only becomes competitive later in training.

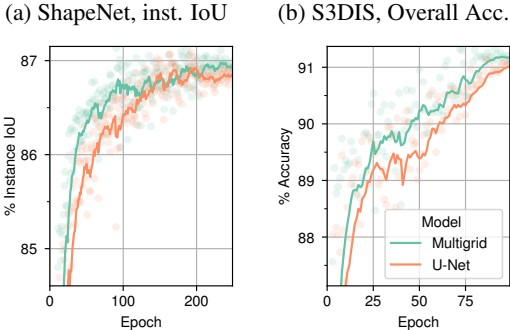

Figure 6: Convergence plots over the course of training on different datasets. We observe that the multigrid network converges faster than the U-Net in the early epochs.

## A.2  ROBUSTNESS AGAINST EDGE SELECTION

In a CNN, the primary way of increasing receptive field size is to increase the size of the kernel. In a graph convolutional network, the equivalent is to increase the number of neighbors, $k$. Section 4.1 indicated that GMNN has an architectural advantage in receptive field size, independent of $k$. Table 7 shows how the performance of DeLA and our multigrid adaptation compare when $k$ is modified. At $k = 20$ (the original setting), both models perform near their best. As $k$ decreases below 16, the accuracy of DeLA falls off rapidly. The multigrid model is much more robust and maintains an accuracy closer to its best accuracy, all the way to $k = 8$. This suggests that the improved receptive field translates to an improved robustness to under-connected networks. More surprisingly, the accuracy of DeLA also falls off when $k$ is increased. This may be due to a combination of overfitting and a missing ability to reject spurious edges. The multigrid network is similarly robust against this type of problem, possibly because it does not have the same dependency on spurious edges for longer-range communication.

Table 7: Effects of k-nearest neighbors on ShapeNet, 25 epochs, 3 trials.

| Model | Accuracy % (mIoU, mean ± std) | | | | | | |
|---|---|---|---|---|---|---|---|
| | $k = 8$ | $k = 12$ | $k = 16$ | $k = 20$ | $k = 24$ | $k = 28$ | $k = 32$ |
| DeLA | 84.1 ± 0.2 | 84.3 ± 0.1 | 84.7 ± 0.2 | 84.8 ± 0.0 | 84.7 ± 0.2 | 84.6 ± 0.1 | 84.6 ± 0.1 |
| GMNN | 84.8 ± 0.2 | 84.9 ± 0.1 | 85.0 ± 0.1 | 85.0 ± 0.0 | 85.0 ± 0.1 | 84.9 ± 0.3 | 84.8 ± 0.1 |

