# OpenReview forum: "Geometric Multigrid Neural Networks"
_ICLR.cc/2026/Conference — ICLR 2026 Conference Withdrawn Submission_

### Official Review · Reviewer_5Pf1 · 2025-10-29

**Soundness:** 2
**Presentation:** 2
**Contribution:** 3
**Rating:** 4
**Confidence:** 4

**Summary:**

This paper introduces Geometric Multigrid Neural Networks (GMNN), a novel architecture for geometric deep learning designed to overcome the fixed information pathways of standard U-Nets. The core contribution is the Geometric Multigrid Convolution (GMC) block, which processes features on all levels of a multigrid hierarchy in parallel and enables flexible, all-to-all communication between scales within each block. Experiments demonstrate its effectiveness.

**Strengths:**

1.	The GMC block is a well-designed and powerful concept. By performing parallel, unshared convolutions on all levels and using efficient progressive transfers for full inter-scale communication, the GMNN creates a highly flexible information routing system.
2.	Extensive experimentation and ablation studies validate the effectiveness of the proposed methods and design choices.
3.	This paper is organized well.

**Weaknesses:**

1.	While the model converges in fewer epochs，this benefit is negated by a significant increase in wall-clock time per epoch. Section 4.6 states that each epoch is 42-62% slower than the baseline, resulting in a total training time that is 14-20% slower. This is a critical trade-off that is obscured in the abstract and introduction, giving a misleading impression of the method's practical efficiency.
2.	The paper claims its architectural contribution is orthogonal to the convolution operator and suggests it could be combined with other layers, such as Transformers. However, all experiments are strictly limited to two specific operators: PointNet++-style and DeLA-style. It will be interesting to provide evidence or experiments to demonstrate that the GMC block, as a general-purpose module, can be directly migrated to other transformer-based baselines (e.g., Point Transformer).
3.	The paper does not provide any qualitative analysis or visualization of the features learned by its core mechanism (the multigrid hierarchy). The existing visualizations are limited to receptive fields (Figure 3) and error maps (Figure 4), which fail to reveal how the GMNN internally organizes and routes semantic information to arrive at its final decision.

**Questions:**

Refer to the Weakness.

---

### Official Review · Reviewer_pGiQ · 2025-10-29

**Soundness:** 2
**Presentation:** 2
**Contribution:** 2
**Rating:** 2
**Confidence:** 3

**Summary:**

This paper proposes a Geometric Multigrid Neural Network for point cloud processing. The main idea is to process multiple scales during each layer, and use down/up-sampling operations to connect different scales. The idea is straightforward and achieves promising performance with a small number of network parameters.

**Strengths:**

- The discussion about connecting different scales is interesting.
- The proposed method achieves promising performance with a small number of network parameters.

**Weaknesses:**

- This paper largely adopts existing concepts in image understanding without introducing novel design considerations for point clouds.
- The method section is inadequately described, omitting critical details necessary for readers to fully comprehend the proposed network. It seems like the core module GMC only contains pointnet-like MLPs for point clouds with different scales and simple down/up-sampling operations for connecting these features.
- A more comprehensive evaluation on large-scale scene datasets (e.g., ScanNet) is required.
- Without a hierarchical design, what is the impact on inference speed when handling large-scale point clouds?

**Questions:**

- How does it perform when replacing the pointnet-like MLP with transformers?

---

### Official Review · Reviewer_EE4k · 2025-10-31

**Soundness:** 2
**Presentation:** 2
**Contribution:** 3
**Rating:** 4
**Confidence:** 4

**Summary:**

This paper proposed a geometric multigrid neural network (GMNN), to address the long-distant feature learning problems in point cloud. Its core is a geometric multigrid convolution blocks which can be stacked to form a deep GMNN. A given point cloud is first represented in pyramid representation and the convolution is occurred among different levels. The experiments demonstrate it can obtain similar accuracy of the base network while using fewer epochs and half the parameter count.

I will decide later whether this paper should be accepted or rejected since 1) I still have some questions which may be answered during rebuttal phase; 2) the idea seems simple and work well on existing deep learning models.

**Strengths:**

1. The idea is simple and its value is verified by experiments.
2. The proposed geometric multigrid convolution block has the potential to combine with existing excellent learning models, just like U-Net.

**Weaknesses:**

1. The idea should be explained in a clearer way. Taking the GMC block for example, I still have difficulty in working out how the GMC block designed. The statement is much more textual description, not clear as U-Net. It should be technical sound. More question are detailed in the Question section.
2. The technical soundness should be better if the source code can be opened. Then, I can reproduce the effect.
3. The value of GMC should be verified by more diversified existing models, not only DeLA. In such a way, it is more convincing.

**Questions:**

1. The multigrid feature description in section 3.1 is clear. it is too short to be an independent subsection.
2. I have some difficulties to figure out how to do the cross-level convolution. It is better to use some mathematical formula or better figures to illustrate this, similar with U-Net, instead of figure 2 in this manuscript.
3. In section 3.2, why the upsampling and downsampling operations are both S(S-1)/2. How to perform progressively, reducing the operations to S-1? I think this is important to understand why the GMC can obtain similar results with less epochs.
4. Why the GMNN can obtain similar accuracy with half parameters? I can not connect this to the GMC block structure. This need clearer explanation and deep analysis.

---

### Official Review · Reviewer_wtQq · 2025-11-01

**Soundness:** 2
**Presentation:** 3
**Contribution:** 2
**Rating:** 4
**Confidence:** 4

**Summary:**

This paper introduces an architecture for geometric deep learning on point clouds. Specifically, it incorporates multigrid feature hierarchies into the network design, enabling communication across different spatial scales through Geometric Multigrid Convolution (GMC) blocks. Each GMC operates on different levels of the hierarchy in parallel, facilitating long-range information propagation and flexible feature routing between coarse and fine representations. The effectiveness of the proposed method is demonstrated on several tasks on 3D point clouds.

**Strengths:**

* The paper is clearly written and well-organized, and it is not hard for a graduate student to implement.

* The design idea is intuitive, i.e., integrating multigrid-style feature representations into point-based networks to enhance multi-scale communication.

* Ablation studies are conducted to analyze the effectiveness of each design.

* The model achieves similar or slightly better performance with fewer parameters.

**Weaknesses:**

* The concept of multigrid convolutional architectures is not new to me. Previous works such as Multigrid Neural Architectures like MG-GNN (Taghibakhshi et al., 2023) already explore similar ideas in other domains. The current work mainly extends these ideas to point clouds, which I think is not novel enough for a top conference.
* The comparisons focus primarily on PointNet++, KPConv, and U-Net variants, but lack evaluations against more recent transformer-based or diffusion-based methods.
* Although the proposed GMNN is presented as a general geometric backbone, the experiments are almost limited to segmentation tasks (ShapeNet-Part and S3DIS). No evaluations are provided on other core 3D learning problems such as classification and surface reconstruction, where multi-scale communication would be particularly relevant.
* The reported accuracy gains are marginal, as shown in Tables 4 and 5.

**Questions:**

Please refer to the weaknesses part. I am open to being persuaded based on the feedback from the authors.

---

### Note · Authors · 2025-11-20

**Comment:**

Thank you for the constructive feedback, we will use it to improve our paper.

**Withdrawal Confirmation:**

I have read and agree with the venue's withdrawal policy on behalf of myself and my co-authors.